# Modelling the emergence of whisker barrels

Sebastian S James[1]*, Leah A Krubitzer[2], Stuart P Wilson[1]

[1]Department of Psychology, The University of Sheffield, Sheffield, United Kingdom;
[2]Center for Neuroscience, The University of California, Davis, Davis, United States

**Abstract** Brain development relies on an interplay between genetic specification and self-organization. Striking examples of this relationship can be found in the somatosensory brainstem, thalamus, and cortex of rats and mice, where the arrangement of the facial whiskers is preserved in the arrangement of cell aggregates to form precise somatotopic maps. We show in simulation how realistic whisker maps can self-organize, by assuming that information is exchanged between adjacent cells only, under the guidance of gene expression gradients. The resulting model provides a simple account of how patterns of gene expression can constrain spontaneous pattern formation to faithfully reproduce functional maps in subsequent brain structures.

## Introduction

Spatial patterns in neural connectivity provide clues about the constraints under which brains evolve and develop (***Purves et al., 1992***). Perhaps the most distinctive pattern can be found in the barrel cortex of many rodent species (***Woolsey and Van der Loos, 1970***). The barrels are identifiable soon after birth in layer 4 of primary somatosensory cortex as dense clusters of thalamocortical axons, which are enclosed by borders a few neurons thick from postnatal day 3 (***Erzurumlu and Gaspar, 2012***). In the plane tangential to the cortical surface the barrels constitute a somatotopic map of the whiskers, with cells within adjacent barrels responding most strongly and quickly to deflection of adjacent whiskers (***Armstrong-James et al., 1992***). Barrel patterning reflects subcortical whisker maps comprising cell aggregates called barrelettes in the brainstem and barreloids in the thalamus (***Ma, 1991***; ***Van Der Loos, 1976***).

Barrel formation requires afferent input from whisker stimulation and thalamic calcium waves (***Antón-Bolaños et al., 2019***), and depends on a complex network of axon guidance molecules such as ephrin-A5 and A7 and adhesion molecules such as cadherin-6 and 8 (***Vanderhaeghen et al., 2000***; ***Miller et al., 2006***). This network is orchestrated by interactions between morphogens Fgf8 and Fgf17 and transcription factors Emx2, Pax6, Sp8, and Coup-tf1 (***Shimogori and Grove, 2005***; ***Bishop et al., 2000***), which are expressed in gradients spanning the cortical sheet that mark orthogonal axes and can be manipulated to stretch, shrink, shift, and even duplicate barrels (***Assimacopoulos et al., 2012***).

The barrel boundaries form a Voronoi tessellation (***Senft and Woolsey, 1991***; ***Figure 1A***), suggesting that barreloid topology is preserved in the projection of thalamocortical axons into the cortex, and that a barrel forms by lateral axon branching from an initial centre-point that ceases upon contact with axons branching from adjacent centres. However, the assumption of pre-arranged centre-points is difficult to resolve with the observation that axons arrive in the cortical plate as an undifferentiated bundle, *prior* to barreloid formation (***Agmon et al., 1993***). In mice, axons from the trigeminal ganglion arrive in the principal division of the trigeminal nucleus (PrV) at E12, then axons from the PrV arrive in the ventroposteromedial nucleus of the thalamus (VPM) at E17, then axons from the VPM arrive in the cortical plate at E18/P0. Distinct whisker-related clusters then become

*For correspondence:
seb.james@sheffield.ac.uk

**Competing interests:** The authors declare that no competing interests exist.

**eLife digest** How does the brain wire itself up? One possibility is that a precise genetic blueprint tells every brain cell explicitly how it should be connected to other cells. Another option is that complex patterns emerge from relatively simple interactions between growing cells, which are more loosely controlled by genetic instruction.

The barrel cortex in the brains of rats and mice features one of the most distinctive wiring patterns. There, cylindrical clusters of cells – or barrels – are arranged in a pattern that closely matches the arrangement of the whiskers on the face. Neurons in a barrel become active when the corresponding whisker is stimulated. This precise mapping between individual whiskers and their brain counterparts makes the whisker-barrel system ideal for studying brain wiring.

Guidance fields are a way the brain can create cell networks with wiring patterns like the barrels. In this case, genetic instructions help to create gradients of proteins across the brain. These help the axons that connect neurons together to grow in the right direction, by navigating towards regions of higher or lower concentrations. A large number of guidance fields could map out a set of centre-point locations for axons to grow towards, ensuring the correct barrel arrangement. However, there are too few known guidance fields to explain how the barrel cortex could form by this kind of genetic instruction alone.

Here, James et al. tried to find a mechanism that could create the structure of the barrel cortex, relying only on two simple guidance fields. Indeed, two guidance fields should be enough to form a coordinate system on the surface of the cortex. In particular, it was examined whether the cortical barrel map could reliably self-organize without a full genetic blueprint pre-specifying the barrel centre-points in the cortex.

To do so, James et al. leveraged a mathematical model to create computer simulations; these showed that only two guidance fields are required to reproduce the map. However, this was only the case if axons related to different whiskers competed strongly for space while making connections, causing them to concentrate into whisker-specific clusters. The simulations also revealed that the target tissue does not need to specify centre-points if, instead, the origin tissue directs how strongly the axons should respond to the guidance fields. So this model describes a simple way that specific structures can be copied across the central nervous system.

Understanding the way the barrel cortex is set up could help to grasp how healthy brains develop, how brain development differs in certain neurodevelopmental disorders, and how brain wiring reorganizes itself in different contexts, for example after a stroke. Computational models also have the potential to reduce the amount of animal experimentation required to understand how brains are wired, and to cast light on how brain wiring is shaped by evolution.

---

apparent in the PrV at P0-P1, in the VPM at P2-P3, and in the cortex at P3-P5 (*Erzurumlu and Gaspar, 2012*; *Sehara and Kawasaki, 2011*).

Alternatively, reaction-diffusion dynamics could generate a Voronoi tessellation without pre-arranged centres, by amplifying characteristic modes in a noisy initial distribution of axon branches, as a net effect of short-range cooperative and long-range competitive interactions. Accordingly, the barrel pattern would be determined by the relative strength of these interactions and by the shape of the cortical field boundary. However, intrinsic cortical dynamics alone cannot account for the topographic correspondence between thalamic and cortical domains, the irregular sizes and specific arrangement of the barrels in rows and arcs, or the influence of gene expression gradients.

The centre-point and reaction-diffusion models are not mutually exclusive. Pre-organized centres could bias reaction-diffusion processes to generate specific arrangements more reliably, and mechanisms of lateral axon branching may constitute the tension between cooperation and competition required for self-organization. However, proof that barrel patterning can emerge from an undifferentiated bundle of axons, based only on local interactions, would show that a separate stage and/or extrinsic mechanism for pre-organizing thalamocortical connections need not be assumed. To this end, we ask whether barrel maps can emerge in a system with reaction-diffusion dynamics, under the guidance of signalling gradients, and in the absence of pre-defined centres.

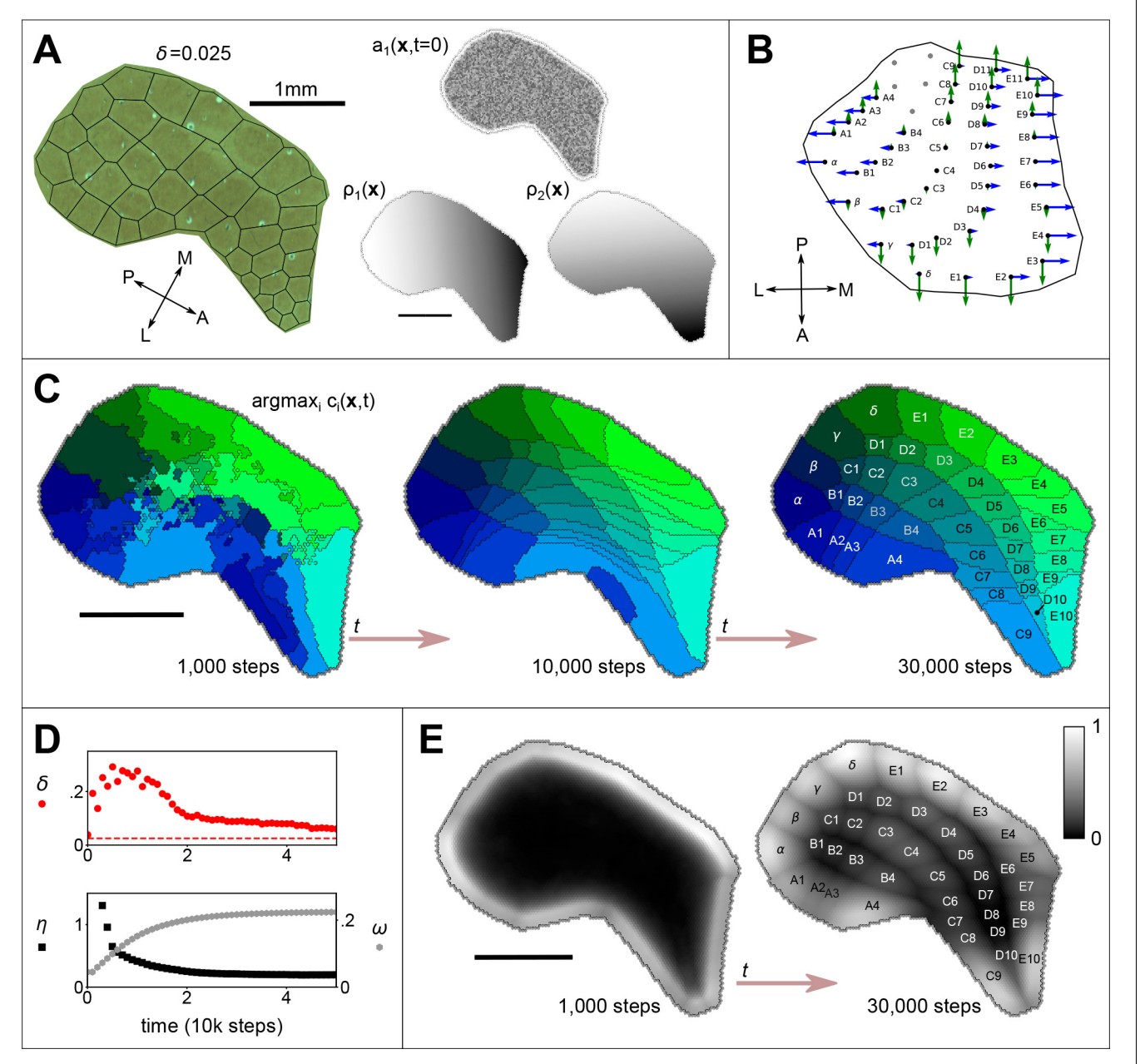

**Figure 1.** The emergence of whisker barrels. (A) Left shows a cytochrome oxidase (CO) stain obtained from rat S1 by **Zheng et al., 2001**, with black lines to delineate barrels and to measure departure (Honda-δ; see **Senft and Woolsey, 1991**) from a perfect Voronoi tessellation. Right shows the initial distribution of axon branching density (a) for one thalamocortical projection, and two molecular guidance fields (ρ), where the domain $S$ has been traced from the CO stain. (B) The strengths of interaction γ with fields $\rho_1$ and $\rho_2$ are indicated for each of 41 projections by the lengths of green and blue arrows respectively, assuming that similar fields aligned to the posterior-anterior and medial-lateral axes in the ventroposterior medial nucleus of the thalamus are sampled at the locations of putative barreloid centres (reconstructed from **Haidarliu and Ahissar, 2001**, their Figure 5b). (C) Results for the example simulation, with parameters $N = 41$, $\alpha = 3.6$, $\beta = 16.67$, $k = 3$, $D = 0.5$, $\gamma \in \pm 2$, $\epsilon = 1.2$ and $\delta t = 0.0001$. Colours indicate the thalamic projection for which the connection density is maximal, barrel labels are located at the centroid of each region and black lines delineate boundaries (see **Figure 1—video 1**). (D) Red dots show the *Honda–δ(t)* metric obtained from the simulation approaching that obtained from the real barrels in A (dotted line); black squares show the *pattern difference* metric $\eta(t)$, and reveal the emergence of a correspondence between the real and simulated barrel shapes (units mm³); grey hexagons show how selectively each cortical site is innervated; $\omega(t) = \iint_S \mu(\mathbf{x}, t)\mathrm{d}S$, where $\mu(\mathbf{x}) \equiv \max_i(c_i(\mathbf{x}, t)) / \sum_{j=1}^{N} c_j(\mathbf{x}, t)$. (E) Plotted across the cortical sheet, the selectivity develops to reveal an alignment with the emergent barrel boundary shapes. Greyscale colour indicates values of $\mu(\mathbf{x})$. All scale bars 1 mm.

The online version of this article includes the following video for figure 1:

*Figure 1 continued on next page*

*Figure 1 continued*

**Figure 1—video 1.** Video corresponding to *Figure 1C* in the main paper.

https://elifesciences.org/articles/55588#fig1video1

## Models

*Karbowski and Ermentrout, 2004* developed a reaction-diffusion style model of how extrinsic signalling gradients can constrain the emergence of distinct fields from intrinsic cortical dynamics. Their model defines how the fraction of occupied synapses $c_i(x,t)$ and the density of axon branches $a_i(x,t)$ interact at time $t$, along a 1D anterior-posterior axis $x$, for *N* thalamocortical projections indexed by *i*. The model was derived from the assumption that the rates at which $a_i$ and $c_i$ grow are reciprocally coupled. Extending the original 1D model to simulate arealization on a 2D cortical sheet, we use $a_i(\mathbf{x},t)$ and $c_i(\mathbf{x},t)$, and model synaptogenesis as

$$\frac{\partial c_i}{\partial t} = -\alpha c_i + \beta \left(1 - \sum_{j=1}^{N} c_j\right)[a_i]^k. \tag{1}$$

Accordingly, where the total fraction of synaptic connections sums to one, connections decay at rate α. Otherwise, $c_i(\mathbf{x},t)$ increases non-linearly (*k*>1) with the density of axon branching. Axon branching is modelled as

$$\frac{\partial a_i}{\partial t} = \nabla \cdot \left(D\nabla a_i - a_i \sum_{j=1}^{M} \gamma_{i,j}\nabla \rho_j(\mathbf{x}) + \chi_i\right) - \frac{\partial c_i}{\partial t}. \tag{2}$$

The first term on the right describes the divergence (indicated by $\nabla\cdot$) of the quantity in parentheses, which is referred to as the 'flux' of axonal branching. The flux represents diffusion across the cortical sheet, at rate *D*, and the influence of *M* molecular signalling fields, $\rho(\mathbf{x})$. The influence of a given field (indexed by *j*) on a given thalamic projection (indexed by *i*), is determined by $\gamma_{i,j}$, which may be positive or negative in order that axons may branch in the direction of either higher or lower concentrations. Note that computing the divergence in simulation requires cells on the cortical sheet to communicate with immediately adjacent cells only (see Methods). Here $\chi_i = 0$ is a placeholder. The second term on the right represents the coupling between axon branching and synaptogenesis, and an assumption that the spatial distribution of synaptic density across the cortical sheet is broadly homogeneous. As such, the quantity $c_i$ can be thought of as the connection density.

## Results

First, we verified that all results established by *Karbowski and Ermentrout, 2004* for a 1D axis could be reproduced using our extension to a 2D cortical sheet. Using an elliptical domain, *S*, with $M = 3$ offset guidance gradients aligned to the longer axis, $N = 5$ thalamocortical projections gave rise to five distinct cortical fields at locations that preserved the topographic ordering defined by the original γ values. However, we found that specifying *N* ordered areas required $M \approx (N + 1)/2$ signalling fields. This is because localization of axon densities occurs only when projections are influenced by interactions with two or more signalling gradients that encourage migration in opposing directions. As the number of guidance fields is unlikely to approach the number of individual barrels, modifications to the model were required.

We reasoned that an arbitrary number of distinct field locations may be determined by a minimum of two guidance gradients, if the concentration of the projection densities is influenced by competition between projections, and if a projection that interacts more strongly with a given guidance gradient migrates further in the direction of that gradient. Accordingly, projections that interact most strongly with a given guidance gradient would come to occupy cortical locations at which that field has extreme values, leaving adjacent locations available to be occupied by projections with the next strongest interactions, and so forth. This would in principle allow the *relative* locations of the fields to be specified by the relative values of the interaction parameters, γ, and hence for a topological map in the cortex to be specified by a spatial ordering of the γ values at the level of the thalamus.

Such dynamics are quite unlike those described by classic chemospecificity models (*Sperry, 1963*), which essentially assume centre-points by specifying conditions in the target tissue that instruct pre-identified afferents to stop growing. Consider, for example, that when simulated in isolation from one-another, all projections in the model described would simply migrate to the extrema of the cortical guidance fields.

Testing this reasoning required increasing the strength of the competition between simulated thalamocortical projections for cortical territory, by increasing the tendency for each projection to compete for cortical space in which to branch and make connections. The major modification required was thus to introduce into the model an additional source of competition between thalamic projections. The term in parentheses in *Equation 1* represents competition between thalamocortical projections for a limited availability of cortical connections. To introduce competition also in terms of axon branching, whilst ensuring that $a_i$ is conserved over time, we redefined

$$\chi_i(\mathbf{x}, t) = \frac{\epsilon a_i}{N-1} \nabla \sum_{j \neq i}^{N} a_j. \tag{3}$$

This term contributes to the *flux* of axonal branching as an additional source of diffusion, scaled by $\epsilon$, which reduces the branching density for a given projection where the branches of other projections are dense. Note that this operation is local to individual afferent projections.

In addition, the model we have outlined requires that molecular guidance gradients in the cortex are complemented by graded values of the interaction strengths, γ, at the level of the thalamus. While the precise mechanisms by which thalamic and cortical gradients interact during development have not been fully characterised, the presence of complementary thalamic and cortical guidance gradients has been well established experimentally. In particular, the EphA4 receptor and its ligand ephrin-A5 are distributed in complementary gradients in the somatosensory thalamus and cortex (*Vanderhaeghen et al., 2000*; *Miller et al., 2006*). Cells originating in VPM express high levels of EphA receptors and project to the lateral part of S1, which expresses low levels of ephrin-A5, and cells originating in the VPL express low levels of EphA receptors and project to the medial part of S1, which expresses high levels of ephrin-A5 (see *Gao et al., 1998*; *Dufour et al., 2003*; *Vanderhaeghen and Polleux, 2004*; *Speer and Chapman, 2005*; *Torii et al., 2013*). We assume that such patterning arises because the relative strengths of interaction with guidance molecules (e. g., ephrin-A5) in the cortex are correlated with the relative concentrations of complementary molecules (e.g., EphA4) in the thalamus, and thus with thalamic position along the axis to which their gradients are aligned.

For simplicity, the two simulated thalamic interaction gradients, as well as the two cortical guidance gradients, were initially chosen to be linear and orthogonal. Hence a given pair of γ values corresponds to the coordinate of a barreloid centre in the VPM. Coordinates, in a reference plane defined by the anterior-posterior and medial-lateral axes, were estimated from Figure 5d of *Haidarliu and Ahissar, 2001*, and scaled such that $\gamma \in \pm 2$. Note that this scaling is arbitrary because according to the model the coordinates provide relative position information only.

A cortical boundary enclosing barrels for 41 macrovibrissae was traced from a cytochrome oxidase stain from *Zheng et al., 2001* (using original data kindly supplied by the authors), and *Equations 1–3* were solved for $N = 41$ projections on the resulting domain, *S*, using $M = 2$ linear signalling gradients aligned with the anterior-posterior and medial-lateral axes. These gradients are shown with the barrel field boundary in *Figure 1A* for clarity, though like ephrin-A5 they may be thought of as extending across the cortical hemisphere (*Miller et al., 2006*). Simulations were stepped through 30000 iterations of *Equations 1–3* ($\delta t = 0.0001$).

Across a wide range of parameter values, random initial conditions (a uniform random distribution for $a(\mathbf{x}, 0) \in (0.2, 0.4)$, $c(\mathbf{x}, 0) = 0$) eventually yielded a clear Voronoi-like tessellation of topographically organized thalamocortical projections, confirming that barrel maps can self-organize in the absence of pre-specified centre points. The organization is apparent in a plot of the identity of the projection for which the connection density is maximal at each simulated cortical location, as shown in *Figure 1C*. Parameter values for this example simulation (see also *Figure 1—video 1*) were obtained by conducting a full parameter sweep and choosing a combination ($\alpha = 3.6$, $\beta = 16.67$, $k = 3$, $D = 0.5$, $\epsilon = 1.2$) that scored well against the following three measures.

First, we used an algorithm introduced by Honda to measure the discrepancy of each barrel shape from a Dirichlet domain shape (*Honda, 1983*). Low overall values of this *Honda-δ* metric

obtained from simulated barrels indicate a close correspondence of the simulated barrel field with a Voronoi tessellation, and thus with a biological barrel field (for mice $\delta \approx 0.054$, *Senft and Woolsey, 1991*, and our analysis of data from *Zheng et al., 2001* indicates that the value for rats is similar). For the tessellation that is overlaid on the real barrel field in *Figure 1A*, $\delta = 0.025$, and a reduction in $\delta$ in the example simulation over time confirmed that an equivalent 'good' Voronoi pattern can emerge within $\approx$ 20000 iterations (*Figure 1D*, red circles). Second, we devised a *pattern difference* measure that is sensitive to deviations in the component shapes and overall topographic registration between two tessellations, $\eta$, and we used this measure to compare the simulated barrel fields to the real barrel field from which the boundary shape applied to the simulation was obtained (see Methods for details). A similar reduction in $\eta$ in the development of the example simulation confirmed that the shapes and arrangement of emergent connection fields came to match those of the real barrel field by around 20000 iterations (*Figure 1D*, black squares). Third, we measured the *connection selectivity, $\omega$,* at each location on the cortical sheet, as the connection density of the most dense projection divided by the sum over all projection densities. The overall connection selectivity increased as the barrel map self-organized in the example simulation (*Figure 1D*, grey hexagons), and the selectivity became concentrated in regions overlapping with the emergent barrel centres (*Figure 1E*).

Against these three metrics we are also able to characterise the robustness of self-organization to the model parameters, and to investigate the sensitivity of the model to variation in its inputs. *Figure 2A* shows values of $\delta$, $\eta$ and $\omega$ obtained after 30000 iterations, from 216 independent simulations, each representing a unique combination of the model parameters $D$, $\epsilon$, and the ratio $\alpha/\beta$. First, we observe that self-organization is highly robust to the ratio $\alpha/\beta$, across five orders of magnitude, with respect to all three metrics. Second, the most strongly Voronoi-conforming patterns (low *Honda*-$\delta$) were generated by simulations in which the diffusion constant $D$ and the strength of competition $\epsilon$ were high. Third, strongest overall connection selectivities, $\omega$, were obtained for lower values of $D$. Fourth, variation in the pattern difference metric, $\eta$, indicated that the alignment between real and simulated patterns was greatest for intermediate rates of diffusion, $D \approx 0.5$. Together these results indicate that when competition is strong, the rate of diffusion determines a trade-off such that fields emerge to be barrel-shaped when diffusion is fast and they emerge to be more selectively innervated when diffusion is slow.

The parameters of the example simulation are indicated in *Figure 2A* using an asterisk. In *Figure 2B*, we also present examples of alternative patterns that emerge for different choices of $D$. Decreasing the rate of diffusion may be considered equivalent to increasing the overall size of the domain, *S*. Hence, insights into barrel development in species with a larger representation of the vibrissae, which do not have barrel fields, may be gained by studying pattern formation when $D$ is small. In this context, it is interesting to note that for small $D$, the organization is predicted to be topological but highly irregular, with a general expansion in the territory occupied by the central versus peripheral domains that would presumably manifest as an absence of identifiable barrel fields (*Figure 2Biii*).

Next we conducted a sensitivity analysis to determine the extent to which the quality of the pattern (after $t = 30000$ iterations) is affected by perturbations to (i) the magnitude and offset of the noise applied to $a_i$ at $t = 0$; (ii) noise applied to the interaction parameters, $\gamma_{i,j}$; (iii) noise (at various length scales) applied to the guidance fields; and (iv) the magnitude and orientation of one cortical guidance field relative to the other (*Figure 1A*).

Using the parameters of the example simulation (*Figure 1C*) we established baseline mean and standard deviations from ten independent simulations with initial uniform random values for $a(\mathbf{x}, 0) \in (0.2, 0.4)$, to be $\delta = 0.089 \pm 0.004$, $\eta = 0.2108 \pm 0.002$, and $\omega = 0.2165 \pm 0.0001$. Repeating with the variation in the initial noise doubled ($a(\mathbf{x}, 0) \in (0.1, 0.5)$), or removed altogether ($a(\mathbf{x}, 0) = 0.3$), generated distributions of $\delta$, $\eta$, and $\omega$ that were not statistically different, as established using paired two-sample t-tests. Adding noise to the interaction parameters ($\gamma$) affected neither the *Honda*-$\delta$ or the *connection selectivity* measures substantially (see *Figure 3A*), and an increase in the *pattern difference* reflected an increase in the occurrence of topological defects only when perturbations became so large as to cause the ordering of $\gamma$ values from neighbouring thalamic sites to be switched (see example map *Figure 3Bi*). Adding noise to the cortical guidance field values, $\rho_1(\mathbf{x})$ and $\rho_2(\mathbf{x})$, disrupted pattern formation only for high levels of noise applied at short length scales, which manifested as non-straight edges at the domain boundaries (*Figure 3Bii*).

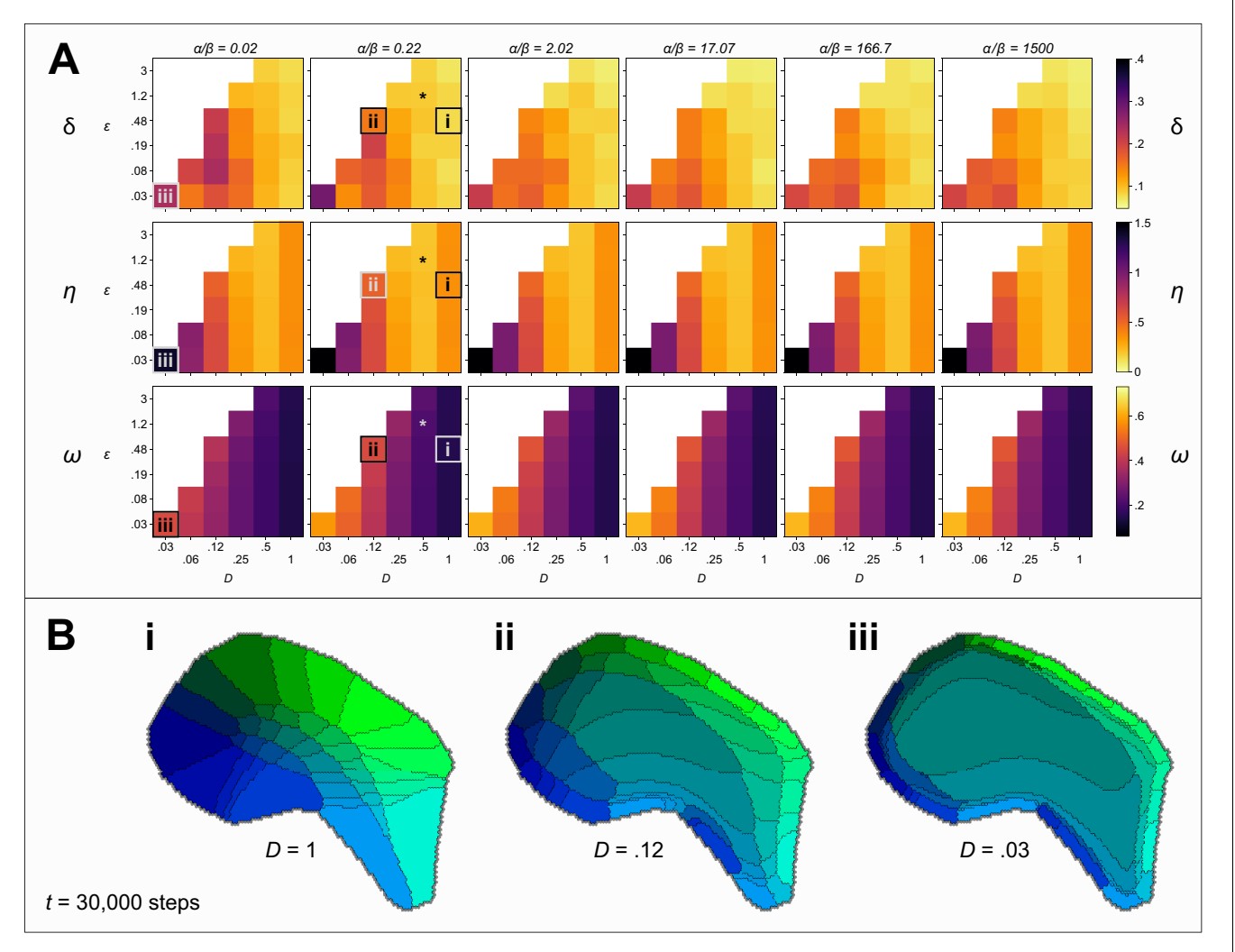

**Figure 2.** Exploring the parameter space. (A) Colour indicates the quality of the pattern at $t = 30000$ steps, against three measures: Low values of *Honda*–δ (top row) suggest a Voronoi-like pattern of fields. The *pattern difference*, η (middle row) measures the difference in the area and arrangement of barrels in real and simulated fields. The *connection selectivity*, ω (bottom row), measures the specificity with which the cortical sheet is innervated. Colour maps are chosen so that lighter (orange and yellow) colours indicate higher quality patterns against each measure. White squares indicate combinations of parameters for which simulations were numerically unstable. The parameter space explored is three dimensional with the ratio $\alpha/\beta$ varying between plots, and the competition parameter $\epsilon$ and the diffusion constant $D$ varying within plots. An asterisk (*) marks the parameters used in *Figure 1*. Boxes (i), (ii) and (iii) mark parameter sets for which corresponding patterns are shown in B (for $t = 30000$ steps). (B) Varying the diffusion constant $D$ generates qualitatively different patterns. Higher values cause expansions of the peripheral barrels and a corresponding compression of the inner barrels (i). Lower values instead cause an expansion of the central barrels and compression of the peripheral barrels (ii). Further reducing the rate of diffusion (iii), which is equivalent to increasing the size of the domain and hence simulating development in an animal with a larger cortex, causes a large area to be occupied by projections with intermediate interaction parameters; those with strong interaction parameters are compressed around the edge of the domain, and consequently a barrel pattern fails to form.

Varying the slope of one linear gradient $\rho_1(\mathbf{x})$ while keeping that of the other constant caused elongation of the emergent domains along the corresponding axis (*Figure 3Biii*), while pattern formation was not strongly influenced by relaxing the assumption that the gradients of the two cortical guidance fields are orthogonal (*Figure 3Biv*). Overall, the sensitivity analysis revealed that self-organization of barrel-like fields in the model is highly robust to a wide range of sources of perturbation.

To further investigate the interplay of genes intrinsic to the developing neocortex and extrinsic factors such as thalamocortical input, we simulated two well known experimental manipulations of barrel development. First, we simulated a seminal barrel duplication paradigm (*Shimogori and*

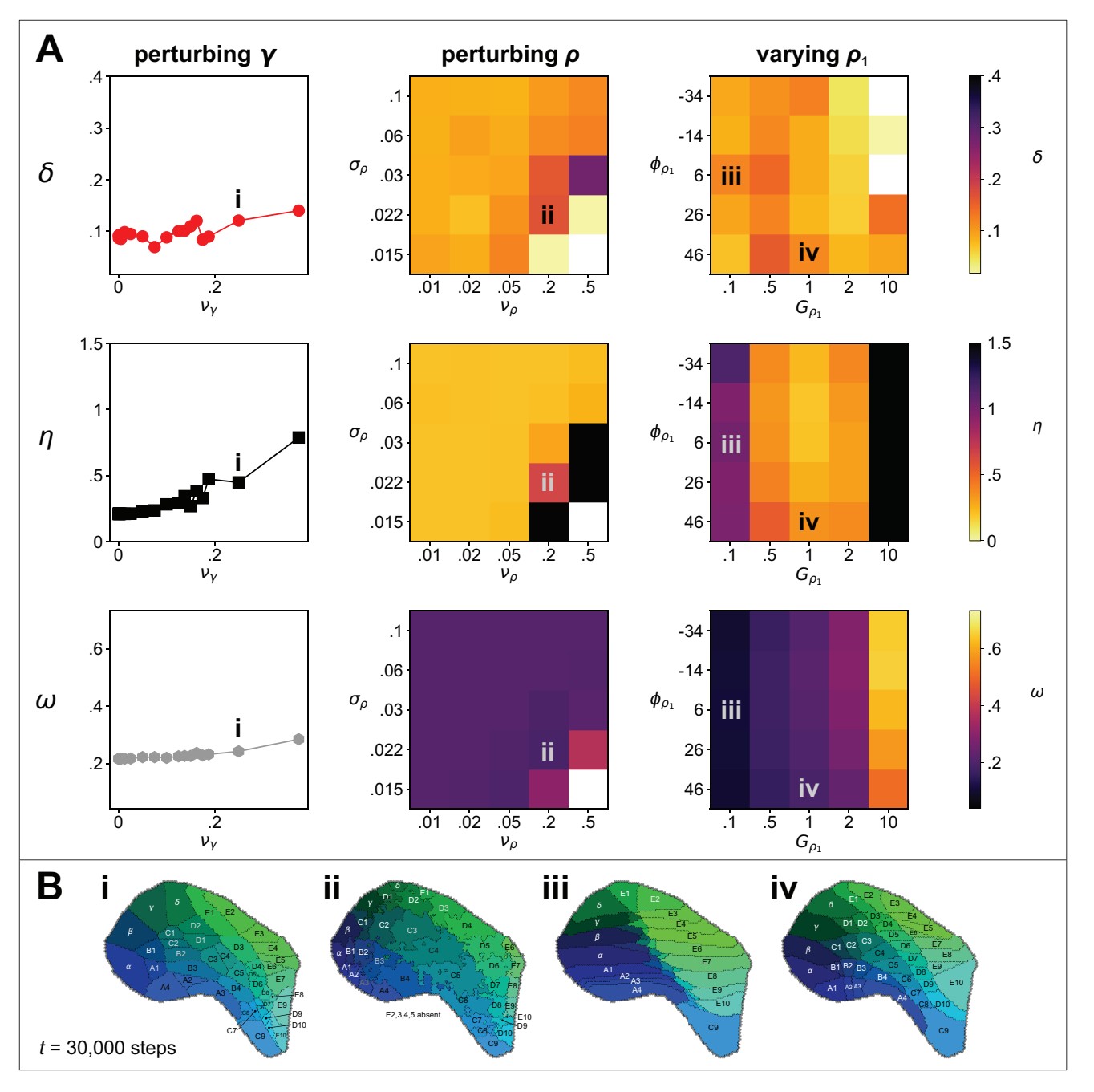

**Figure 3.** Sensitivity analysis. (**A**) Metrics of map quality $\delta(t)$ (top row), $\omega(t)$ (middle row) and $\eta(t)$ (bottom row), were evaluated at $t = 30000$ steps. The $y$-axes and colour scales have identical ranges to the colour scales in *Figure 2*, for easy comparison. *Left column:* The effect of adding noise drawn from a uniform distribution, $(\gamma_{max} - \gamma_{min})\,U(0, \nu_\gamma)$, to the values of the interaction parameters, $\gamma$ used in *Figure 1*. *Middle column:* The effect of changing the magnitude and the length scale of noise applied to the guidance fields. Uniform random noise $(\rho_{max} - \rho_{min})\,U(0, \nu_\rho)$ was added to each (hexagonal) element of $\rho_1(\mathbf{x})$ and $\rho_2(\mathbf{x})$ and the result was smoothed by convolution with a symmetric 2D Gaussian kernel of width $\sigma_\rho$. *Right column:* The effect of setting the rotational angle of the linearly varying guidance field, $\rho_1(\mathbf{x})$, to $\phi_{\rho_1}$, and modifying its overall gain to $G_{\rho_1}$, whilst keeping the parameters of $\rho_2(\mathbf{x})$ unchanged from those used in the example simulation, for which $\phi_{\rho_2} = 84°$ and $G_{\rho_2} = 1$. (**B**) Four ways in which the perturbations in A affect the patterns. (i) The effect of significant interaction parameter noise is to introduce topological defects. (ii) High magnitude, short length scale noise in $\rho(\mathbf{x})$ leads to non-straight edges between adjacent barrels. (iii) Reducing the slope of gradient $\rho_1$ (by a factor of 10) causes barrel rows B, C and D to become 'crushed' down the centre line, and edge barrels to dominate. (iv) Rotating $\rho_1$ by 20° causes a slight distortion of the pattern, resulting in an overall anticlockwise rotation of the field locations.

*Grove, 2005*; *Assimacopoulos et al., 2012*) in which the growth factor Fgf8, which is normally expressed at the anterior end of the cortical subplate from around E9.5 (*Crossley and Martin, 1995*), is ectopically expressed (by electroporation) also at the posterior pole. We assume that this results in a mirror of the primary barrel cortex boundary along the rostrocaudal axis (*Assimacopoulos et al., 2012*) and a mirroring of the anterior-posterior guidance gradient $\rho_1$ at the border between them (*Figure 4A*). The result after 30000 iterations, and otherwise using the parameters of the example simulation, was two mirror-symmetrical barrel fields comprising $2N$ barrels (*Figure 4B*), consistent with the outcome of the original experiments.

Finally, to investigate the response of the model to environmental manipulation, we simulated a whisker deprivation experiment. In a critical period comprising the first postnatal days, removal of the whiskers by electrocauterization, plucking, or trimming leads to observable changes in brain structures, including the barrel field (*Jeanmonod et al., 1981*). Amongst other changes, deprivation of individual whiskers leads to smaller barrels (*Kossut, 1992*). We simulated trimming of the individual whisker C3 during the critical period by reducing the competitiveness of the C3 thalamocortical

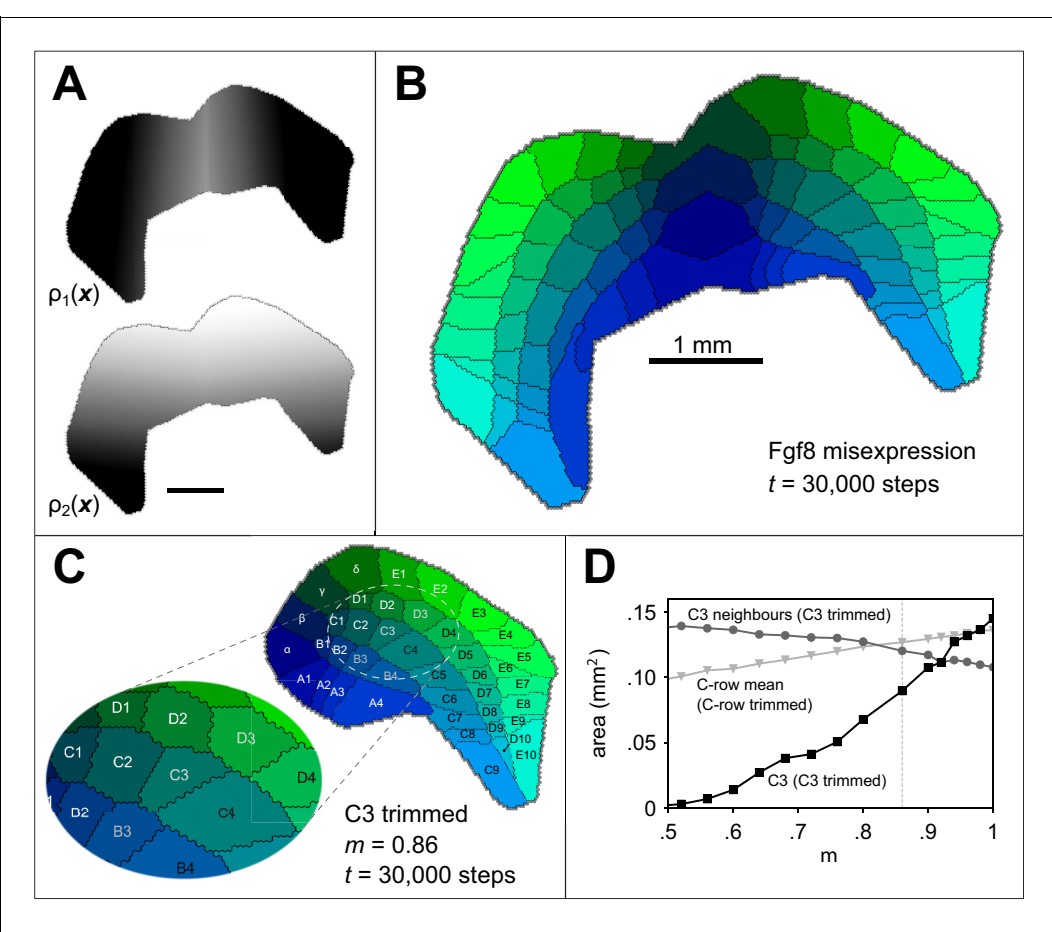

**Figure 4.** Simulating altered barrel development. Guidance fields (**A**) and emergent barrel pattern (**B**) in a Fgf8 misexpression experiment (*Assimacopoulos et al., 2012*), simulated by reflecting $\rho_1$ from *Figure 1A* at the join of the original boundary with its mirror. All other model parameters match those in *Figure 1C*. C Simulating whisker trimming by reducing the competitiveness, $\epsilon$, of one projection. For C3 only, $\epsilon$ was multiplied by $m \in (0, 1)$. The pattern shown is that formed after 30000 steps with $m = 0.86$, which reduces the size of the C3 field to 65% of its original size, matching the average barrel area reduction observed by *Kossut, 1992*. D The area of the C3 barrel (black squares) reduces as $m$ is reduced, whereas the mean area of neighbouring barrels (B3, C2, C4, D2 and D3, grey circles) increases. The dotted grey line indicates $m = 0.86$ for comparison with panel C. If the $\epsilon$ value is instead reduced for *all* row C projections (including the interstitial γ projection), the mean row C barrel area is only slightly reduced (light grey triangles). In this case, the mean area of a simulated row C barrel at $m = 0.86$ is 91% of that for $m = 1$.

projection, $\epsilon$. As a result, the corresponding field size was smaller (*Figure 3C*), and the size of the fields representing the neighbours of C3 increased in size. A reduction in area to 65%, comparable to that induced by *Kossut, 1992*, was obtained in simulation when $\epsilon$ was reduced to 86% of its default value (*Figure 3D*), and the C3 barrel disappeared altogether when $\epsilon$ was less than half of its original value.

Although individual whisker trimming reduces barrel size, if an entire row is trimmed, barrel sizes for the trimmed row are not obviously changed (*Land and Simons, 1985*). We investigated with a simulation in which we varied $\epsilon$ for all of the row C projections. Although the barrels which formed for row C did show some reduction in area (*Figure 4D*), this reduction was small when compared with the individually trimmed C3 simulation. If the effect of trimming any whisker is to reduce its $\epsilon$ (the competitiveness of its projection) to 86% of its original value, then the model predicts that the cortical barrels for the trimmed C row will retain 91% of their area on average.

Together with the results of simulated misexpression, the consistency of the simulated whisker trimming results with those of the original studies demonstrates how the model can be used to investigate the contribution of intrinsic and extrinsic factors to the development of cortical fields.

## Discussion

The present results suggest that the key requirements for the emergence of realistic barrel patterning are (i) at each cortical location thalamocortical projections compete for a limited number of available synaptic connections (*Equations 1–2*), (ii) at each location the branching rate of a given projection is reduced by the density of other projections (*Equation 3*), and (iii) the branch density of each projection is conserved over time.

The emergence of barrels in simulation required competition between thalamic projections in terms of synaptic connectivity and also competition in terms of cortical space, as represented by $\chi$, with an implicit requirement for a self/other identifier amongst projections. This latter form of competition may account for the absence of barrels in rodents with larger brains, such as capybara, for which competition for space is presumably weaker (*Woolsey et al., 1975*). Hence, irrespective of whether barrels are necessary for adaptive whisker function, the emergence of somatotopically ordered modular structures may be an inevitable consequence of local competition for cortical territory driven by input from an array of discrete sensory organs (*Purves et al., 1992*).

In reality, the Voronoi tessellation is extended by scores of smaller barrels alongside the E-row barrels, which represent the microvibrissae, and presumably form via the same competitive processes. Enforcing here the same boundary condition as used to represent the true edges of the barrel field was necessary to ensure the stability of the simulation, though we acknowledge that this region of the boundary was enforced primarily to keep the number of simulated projections, and hence the overall computational complexity of the simulation, manageable (simulating an extra projection introduces 13030 new dynamical variables).

It is important to emphasize that the formulation of the model is entirely local, insofar as simulation requires no information to be communicated from a given cortical grid cell to any but those immediately adjacent (via diffusion). Hence the simulations demonstrate how a self-organizing system, constrained by genetically specified guidance cues and by the shape of the cortical field boundary, can faithfully reproduce an arrangement of cell aggregates in one neural structure as a topographic map in another.

Moreover, the present results confirm that somatotopic map formation does not require the prespecification of centre-points by as yet undetermined additional developmental mechanisms.

## Methods

We concentrated on the representation of the forty-one macrovibrissae that constitute a given barrel field, because their thalamic and cortical correlates are easily identifiable and consistently located, excluding the five rhinal whiskers as their cortical representation is isolated from the main barrel field. We excluded the representation of the microvibrissae to limit the overall complexity of the simulations.

The cortical sheet was modelled as a two dimensional hexagonal lattice, which simplifies the computation of the 2D Laplacian. Within a boundary traced around the edge of a rat barrel field

(*Figure 1A*) we set the hex-to-hex distance $d$ to 0.03 mm, which resulted in a lattice containing 6515 hexes for the simulations shown in *Figure 1A,C and D* and 12739 hexes for the Fgf8 misexpression study shown in *Figure 3*. Each hex contained 82 time-dependent variables: 41 branching densities ($a_i$) and 41 connection densities ($c_i$). The rate of change of each of the time-dependent variables (*Equations 1 and 2*) was computed using a fourth-order Runge-Kutta method.

The most involved part of this computation is to find the divergence of the flux of axonal branching, $\mathbf{J}_i(\mathbf{x},t)$, the term in parentheses in *Equation 2*:

$$\nabla \cdot \mathbf{J}_i(\mathbf{x},t) = \nabla \cdot \left( D\nabla a_i - a_i \sum_{j=1}^{M} \gamma_{i,j} \nabla \rho_j(\mathbf{x}) + \frac{\epsilon a_i}{N-1} \nabla \hat{a}_i \right), \tag{4}$$

where $\hat{a}_i \equiv \sum_{j \neq i}^{N} a_j$. Note that the sum of the guidance gradients is time-independent and define $\mathbf{g}_i(\mathbf{x}) \equiv \sum_{j=1}^{M} \gamma_{i,j} \nabla \rho_j(\mathbf{x})$. Because the divergence operator is distributive, *Equation 4* can be expanded using vector calculus identities (dropping references to $\mathbf{x}$ and $t$ for clarity):

$$\nabla \cdot \mathbf{J}_i = \nabla \cdot (D\nabla a_i) - \nabla \cdot (a_i \mathbf{g}_i) + \frac{\epsilon}{N-1} \nabla \cdot (a_i \nabla \hat{a}_i). \tag{5}$$

Applying the vector calculus product rule identity yields

$$\nabla \cdot \mathbf{J}_i = D\nabla \cdot \nabla a_i - a_i \nabla \cdot \mathbf{g}_i - \mathbf{g}_i \cdot \nabla a_i + \frac{\epsilon a_i}{N-1} \nabla \cdot \nabla \hat{a}_i + \frac{\epsilon}{N-1} \nabla \hat{a}_i \cdot \nabla a_i, \tag{6}$$

which has five elements to compute: (i) $D\nabla \cdot \nabla a_i$ (the Laplacian of $a_i$); (ii) a time-independent modulator of $a_i$ (because $\nabla \cdot \mathbf{g}_i$ is a time-independent static field); (iii) the scalar product of the static vector field $\mathbf{g}_i$ and the gradient of $a_i$; (iv) the Laplacian of $\hat{a}_i$; and (v) a term involving the gradients of $a_i$ and $\hat{a}_i$. Each of the divergences can be simplified by means of Gauss's Theorem following *Lee et al., 2014*.

i. The computation of the mean value of the Laplacian across one hexagon of area $\Omega = \frac{\sqrt{3}}{2}d^2$, located at position $\mathbf{p}_0$, with neighbours at positions $\mathbf{p}_1 - \mathbf{p}_6$ is

$$\begin{aligned} \langle D\nabla \cdot \nabla a_i(\mathbf{p}_0,t) \rangle &= \frac{D}{\Omega} \oiint_\Omega \nabla \cdot \nabla a_i(\mathbf{x},t) \, d\Omega \ = \frac{D}{\Omega} \oint \frac{\partial a_i}{\partial \hat{\mathbf{n}}} d\gamma \\ &\approx \frac{D}{\Omega} \sum_{j=1}^{6} \frac{\partial a_i(\mathbf{p}_j)}{\partial \hat{\mathbf{n}}} {}_{\text{mid}} v \\ &= \frac{2D}{\sqrt{3}d^2} \sum_{j=1}^{6} \frac{a_i(\mathbf{p}_j) - a_i(\mathbf{p}_0)}{d} \frac{d}{\sqrt{3}} \\ &= \frac{2D}{3d^2} \sum_{j=1}^{6} (a_i(\mathbf{p}_j) - a_i(\mathbf{p}_0)), \end{aligned} \tag{7}$$

where $v = d/\sqrt{3}$ is the length of each edge of the hexagon and $d\gamma$ is an infinitesimally small distance along its perimeter.

ii. The computation of the second term in *Equation 6*, $\langle a_i(\mathbf{p}_0,t)\nabla \cdot \mathbf{g}_i(\mathbf{p}_0) \rangle$, can be written out similarly:

$$\begin{aligned} \frac{1}{\Omega} \oiint_\Omega a_i \nabla \cdot \mathbf{g}_i \, d\Omega \quad &= \frac{a_i(\mathbf{p}_0,t)}{\Omega} \oint \mathbf{g}_i \cdot d\hat{\mathbf{n}} \\ &\approx \frac{a_i(\mathbf{p}_0,t)}{\Omega} \sum_{j=1}^{6} \frac{\mathbf{g}_i(\mathbf{p}_j) + \mathbf{g}_i(\mathbf{p}_0)}{2} \cdot \hat{\mathbf{n}} \, v \\ &= \frac{2a_i(\mathbf{p}_0,t)v}{\sqrt{3}d^2} \sum_{j=1}^{6} [\frac{g_i^x(\mathbf{p}_j) + g_i^x(\mathbf{p}_0)}{2} \cdot \hat{\mathbf{n}} + \frac{g_i^y(\mathbf{p}_j) + g_i^y(\mathbf{p}_0)}{2} \cdot \hat{\mathbf{n}}] \\ \Rightarrow \langle a_i(\mathbf{p}_0,t)\nabla \cdot \mathbf{g}(\mathbf{p}_0) \rangle \quad &\approx \frac{a_i(\mathbf{p}_0,t)}{3d} \sum_{j=1}^{6} [(g_i^x(\mathbf{p}_j) + g_i^x(\mathbf{p}_0))\cos(\frac{\pi}{3}(j-1)) + (g_i^y(\mathbf{p}_j) + g_i^y(\mathbf{p}_0))\sin(\frac{\pi}{3}(j-1))], \end{aligned} \tag{8}$$

where $g_i^x$ and $g_i^y$ are the Cartesian components of $\mathbf{g}_i$. Both this last expression, and the final expression of **Equation 7** can be computed locally, by summing over values of the nearest neighbours.

iii. The middle term in **Equation 6** is the scalar product of two vector fields which is straightforward to compute from their Cartesian components.

iv. The same method used to compute $\nabla \cdot \nabla a_i$ in term (i) is used to compute $\nabla \cdot \nabla \hat{a}_i$.

v. The final term is the scalar product of the two vector fields $\nabla a_i$ and $\nabla \hat{a}_i$.

By separating the computation of **Equation 4** into parts (i)–(v), the no-flux boundary condition,

$$\mathbf{J}_i(\mathbf{x}, t)_{\text{boundary}} = 0, \tag{9}$$

can be fulfilled. On the boundary, the contribution to $\mathbf{J}$ resulting from the first term of **Equation 6** can be fixed to 0 by the 'ghost cell method' in which, during the evaluation of (i), a hex outside the boundary containing the same value as the hex inside the boundary is imagined to exist such that the flux of $\mathbf{J}$ across the boundary is 0. Then, $\mathbf{g}_i(\mathbf{x})$ can be tailored so that it, and its normal derivative, approach 0 at the boundary, ensuring that the second and third terms of **Equation 6** also contribute nothing to $\mathbf{J}$. This is achieved by multiplying $\mathbf{g}_i(\mathbf{x})$ by a sharp logistic function of the distance, $d_b$, from $\mathbf{x}$ to the boundary, of the form $1/[1 + \exp(100(d_f - d_b))]$, where $d_f = 0.1$ mm $\approx 3d$ is the boundary fall-off distance.

The *pattern difference* metric, $\eta$, incorporates information about the differences in the areas of simulated and experimentally determined (real) barrels ($\mathcal{A}_i^{\text{sim}}$ and $\mathcal{A}_i^{\text{exp}}$), as well as information contained in the 'adjacency vector' for each barrel, $\mathcal{V}_i$, the $j$-th element of which is the length of the border between barrel $i$ and $j$. For a well-formed barrel pattern, $\mathcal{V}_i$ is a sparse vector. A dimensionless quantity can be obtained from the scalar product of the simulated and experimental adjacency vectors: $\frac{1}{N}\sum_i \frac{\mathcal{V}_i^{\text{sim}}}{b_i^{\text{sim}}} \cdot \frac{\mathcal{V}_i^{\text{exp}}}{b_i^{\text{exp}}}$, where for example $b_i^{\text{exp}}$, is the total length of the border around real barrel $i$. This quantity is small when the fields that form in simulation have dissimilar neighbour relations to those of the real barrels (e.g., **Figure 1C**, $t = 1000$), and maximal for a precise topological map ($t = 10000$). A second comparison considers the mean magnitude of the difference between the simulated and experimental vectors: $\frac{1}{N}\sum_i \mathcal{V}_i^{\text{sim}} - \mathcal{V}_i^{\text{exp}}$. This tends to 0 for a perfect match and can separate patterns with straight boundaries from those with 'noisy edges' (as in **Figure 3Bii**). We combined these terms into a single metric:

$$\eta = \frac{\frac{1}{N}\sum_i \mathcal{A}_i^{\text{sim}} - \mathcal{A}_i^{\text{exp}} \times \frac{1}{N}\sum_i \mathcal{V}_i^{\text{sim}} - \mathcal{V}_i^{\text{exp}}}{\frac{1}{N}\sum_i \frac{\mathcal{V}_i^{\text{sim}}}{b_i^{\text{sim}}} \cdot \frac{\mathcal{V}_i^{\text{exp}}}{b_i^{\text{exp}}}}, \tag{10}$$

which has units of mm$^3$.

The code required to reproduce these results is available at https://github.com/ABRG-Models/BarrelEmerge/tree/eLife (**James, 2020**, copy archived at https://github.com/elifesciences-publications/BarrelEmerge). The computations described in (i)–(v) may be found in the class method RD_James_comp2::compute_divJ() which calculates term1, term2, term3, term1_1 and term1_2, respectively. BarrelEmerge depends on the software library *morphologica* (RRID:SCR_018813), which must also be compiled.

# Acknowledgements

The authors thank Jason Berwick at the University of Sheffield for advice and for access to the rat barrel stains used to construct **Figure 1A**. This work was supported by a Collaborative Activity Award, *Cortical Plasticity Within and Across Lifetimes*, from the James S McDonnell Foundation (grant 220020516).

## Additional information

### Funding

| Funder | Grant reference number | Author |
| --- | --- | --- |
| James S. McDonnell Foundation | 220020516 | Sebastian S James<br>Leah A Krubitzer<br>Stuart P Wilson |

The funders had no role in study design, data collection and interpretation, or the decision to submit the work for publication.

### Author contributions

Sebastian S James, Conceptualization, Software, Formal analysis, Visualization, Writing - original draft; Leah A Krubitzer, Funding acquisition, Writing - review and editing; Stuart P Wilson, Conceptualization, Formal analysis, Funding acquisition, Writing - original draft

### Author ORCIDs

Sebastian S James (iD) https://orcid.org/0000-0003-0208-0588
Stuart P Wilson (iD) https://orcid.org/0000-0001-8125-5133

### Decision letter and Author response

Decision letter https://doi.org/10.7554/eLife.55588.sa1
Author response https://doi.org/10.7554/eLife.55588.sa2

# Additional files

### Supplementary files

• Transparent reporting form

### Data availability

All data and code required to reproduce the results in this work are available in the public repository https://github.com/ABRG-Models/BarrelEmerge/tree/eLife (copy archived at https://github.com/elifesciences-publications/BarrelEmerge).

The following datasets were generated:

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
