## [Decision Letter]

**Acceptance summary:**

The study proposes a compact self-organization model for formation of whisker barrels in the absence of pre-defined centers for the barrels. Further, the authors have come up with a biologically plausible rule for competition and conservation of axonal density.

**Decision letter after peer review:**

Thank you for submitting your article "Modelling the emergence of whisker barrels" for consideration by *eLife*. Your article has been reviewed by three peer reviewers, one of whom is a member of our Board of Reviewing Editors, and the evaluation has been overseen by Timothy Behrens as the Senior Editor. The following individual involved in review of your submission has agreed to reveal their identity: Bard Ermentrout (Reviewer #2).

The reviewers have discussed the reviews with one another and the Reviewing Editor has drafted this decision to help you prepare a revised submission.

As the editors have judged that your manuscript is of interest, but as described below that additional analyses are required before it is published, we would like to draw your attention to changes in our revision policy that we have made in response to COVID-19 (https://elifesciences.org/articles/57162). First, because many researchers have temporarily lost access to the labs, we will give authors as much time as they need to submit revised manuscripts. We are also offering, if you choose, to post the manuscript to bioRxiv (if it is not already there) along with this decision letter and a formal designation that the manuscript is 'in revision at *eLife*'. Please let us know if you would like to pursue this option. (If your work is more suitable for medRxiv, you will need to post the preprint yourself, as the mechanisms for us to do so are still in development.)

Summary:

This compact paper proposes a self-organization model for formation of whisker barrels. The key idea is that reaction-diffusion dynamics can lead to the observed topology, in the absence of pre-defined centers for the barrels.

Essential revisions:

1) How do the authors obtain 41 pairs of γ values (Results, third paragraph)? Are these parameters or were they inferred from experiments? This must be better motivated.

2) The competition term χ*_i_* requires renormalization, which seems biologically implausible. The authors may wish to try a form such asXai∇1N−1∑j≠iajwhich does not need renormalization. Several other points about this competition term are unclear as mentioned in the reviewer comments.

3) There should be more exploration of the model: some parameter exploration and sensitivity analysis, and some more predictions.

Reviewer #1:

This compact paper proposes a self-organization model for formation of whisker barrels. The key idea is that reaction-diffusion dynamics can lead to the observed topology, in the absence of pre-defined centers for the barrels.

The model is well presented and the motivation of mathematical choices is mostly clear. It may be worth expanding on the motivation for competition for axonal branching (Equations 3 and 4).

It is a little unclear how the misexpression experiment (Shimogori and Grove, 2005) in Figure 1E was done. The simulation approach and outcome for this section is described very tersely.

The authors also mention another easily modeled experiment, in which capybara brains lack barrels because they are big. It should be a simple matter to do this run.

Overall I feel this study presents an attractive and compact model for the formation of whisker barrels, which has good biological motivation, and does a good job of reducing assumptions and molecular guidance cues.

Reviewer #2:

This is an interesting paper that with a few assumptions shows that an old model for areal formation in cortex is sufficient to quantitatively reproduce the patterns of barrels observed in mouse S1. It would appear from the model that the key is the parameters *γ_ij_* that are presumably (hypothesized) to be assigned at the level of the thalamus. I have a few questions about the paper.

1) Does the same model work with respect to projections from the brain stem (barrelettes) to the thalamus (barreloids)? This would be a good way to check the ideas. Related to this, is it true that the barrelettes (barreloids) precede the development of the barreloids (barrels)? It would seem to be necessary? Or perhaps, starting with a double gradient in the thalamus and cortex and a prepattern in the barelettes, would the correct patterns emerge simultaneously?

2) There seems to be a strong prediction in this concerning the development of the patterns over time. Figure 1C indicates that early on there are large distortions in the shape of the barrels particularly in Figure 1D, E rows. is this known to occur?

3) It seems to me that without the chi, then possible connections plus axons are conserved which is reasonable. But with the necessary competition, there seems to be a flaw in the model if they have to renormalize at each point. If axons make connections should they not be lost from the pool forever (this is the -dci/dt the model). For example, since the gradient has no flux in the original K&E model, there is conservation of the total number of connections and axons of a given type (int ai+ci dx = constant). This principle seems to make sense to me. However, the competition term χ*_i_* seems to disrupt this. Is there a way to introduce the axonal competition in a way the prevents the unrealistic (or biologically implausible, at least) renormalization at each step? I'd be more comfortable with the model if there were a more physiological way to renormalize. For example, I don’t know if the authors considered something like an additional flux of the form:Xai∇1N−1∑j≠iaj

This makes the axons of type i move away from type j while at the same time enforcing conservation without recourse to some sort of post normalization.

Reviewer #3:

The manuscript studies a theoretical model within the framework of reaction-diffusion equations coupled to signalling gradients to possibly explain the emergence of whisker barrels in the cortex.

1) The model considered by the authors is identical to the one studied by Karbowski and Ermentrout, 2004. The only new features are the extension of the original 1D model to 2D and the addition of an extra term to represent competition in axonal branching.

2) The authors consider 2 guiding fields. What are their explicit spatial profiles? Notice that since these fields essentially guide the emergent pattern and hence their profiles, in relation to the geometry of the 2D domain, are crucial. A different profile would certainly lead to a different pattern. I feel that it is not enough to say '…linear signalling gradients aligned with the anterior-posterior and medial-lateral axes….' since the domain is 2D and of non-rectangular shape.

3) The justification for the introduction of the extra term for competition amongst axons (Equation 3) is missing. Why that form? What is the reasoning for introducing axonal competition? What essential features of the resultant patterns are missed out if this term is absent? Or has a different form? In the Discussion section, the authors mention, without any justification, that the conservation of branch density in each projection is a key requirement for the emergence of barrel patterns. This is totally unclear.

4) Related to the above point, the authors mention that the axonal branch density is bounded by their dynamics. I presume that the integrations on the RHS of Equation 4 are spatial integrals over the domain. Then how come a spatial index survives in the LHS of this equation? How did the authors arrive at this equation? Is there a continuous-time version of this equation (like a conservation law), i.e., one that does not make a reference to the discrete time-stepping dynamics?

5) A typical mathematical modelling study should explore the space of relevant parameters to demonstrate the possible range of behaviours that the model can exhibit. This is usually presented as a phase-diagram. The authors do not explore the parameter space (or the possible spatial profiles of the guiding fields) in their study.

6) Throughout, the authors emphasize the spatial-locality of their mathematical model and conclude 'Hence the simulations demonstrate how a self-organizing system…'. A mathematical model with spatial-locality alone does not imply self-organized dynamics. With a sufficiently large number of spatio-temporal fields (N=42), and the concomitant parameters, and non-autonomous guiding fields, it is possible to reproduce any desired pattern. As such, it is crucial in the mathematical modelling of living systems to delineate the essential requirements from the incidental.

---

## [Author Response]

Essential revisions:1) How do the authors obtain 41 pairs of γ values (Results, third paragraph)? Are these parameters or were they inferred from experiments? This must be better motivated.

We have addressed this comment through edits at several points in the manuscript. At the beginning of the revised Materials and methods section we have clarified why 41 pairs were chosen:

“We concentrated on the representation of the forty-one macrovibrissae that constitute a given barrel field, because their thalamic and cortical correlates are easily identifiable and consistently located, excluding the five rhinal whiskers as their cortical representation is isolated from the main barrel field. We excluded the representation of the microvibrissae to limit the overall complexity of the simulations.”

To the revised Introduction we have added the following motivation:

“We reasoned that an arbitrary number of distinct field locations may be determined by a minimum of two guidance gradients, if the concentration of the projection densities is influenced by competition between projections, and if a projection that interacts more strongly with a given guidance gradient migrates further in the direction of that gradient. […] Consider, for example, that when simulated in isolation from one-another, all projections in the model described would simply migrate to the extrema of the cortical guidance fields.”

Shortly thereafter we have added the following, which identifies the experimental observations on which our inferences are based:

“In addition, the model we have outlined requires that molecular guidance gradients in the cortex are complemented by graded values of the interaction strengths, 𝛾, at the level of the thalamus. […] We assume that such patterning arises because the relative strengths of interaction with guidance molecules (e.g., ephrin-A5) in the cortex are correlated with the relative concentrations of complementary molecules (e.g., EphA4) in the thalamus, and thus with thalamic position along the axis to which their gradients are aligned.”

Finally, in the revised Discussion, we have acknowledged the following:

“In reality, scores of smaller Dirichletform barrels representing the microvibrissae form alongside the E-row barrels, presumably via the same competitive processes. Enforcing here the same boundary condition as used to represent the true edges of the barrel field was necessary to ensure the stability of the simulation, though we acknowledge that this region of the boundary was enforced primarily to keep the number of simulated projections, and hence the overall computational complexity of the simulation, manageable (simulating an extra projection introduces 13030 new dynamical variables).”

Together, we feel that these additions better explain and motivate our assumptions about the interaction parameters in a transparent way.

2) The competition term χ_i_ requires renormalization, which seems biologically implausible. The authors may wish to try a form such asXai∇1N−1∑j≠iajwhich does not need renormalization. Several other points about this competition term are unclear as mentioned in the reviewer comments.

This is a very insightful comment. Addressing it led us to make important modifications to the model, and substantial improvements to the paper. We had previously attempted to use the form for the competition term that is suggested here, but without success. However, to address point 3 below we have since developed new software tools to facilitate a large-scale parameter sweep (and sensitivity analysis), and when we carried out that full parameter sweep using the form for the competition term suggested, we discovered that during our original investigations we had been exploring only a small region of the parameter space in which the dynamics happened to be unstable. Now, in the region of the parameter space in which simulations are numerically stable, we have found pattern formation via the reformulated model to be qualitatively similar and highly robust. We too prefer the suggested reformulation, which removes the need for an explicit re-normalization step (original Equation 4), and in the revised paper all of the results are now obtained using the more elegant form of the model (revised Equation 3). Note that the resulting expression is identical to that suggested, but that it must be included as part of the flux of axonal branching term, hence the placeholder variable 𝜒 is included within the parentheses in Equation 2 of the revision (previously it appeared outside). Note also that concomitant edits have been made to the Materials and methods section to reflect minor changes to the method of numerical integration that was required to realize the revised form of the competition.

3) There should be more exploration of the model: some parameter exploration and sensitivity analysis, and some more predictions.

The revision now includes a report of the results of fully exploring the parameter space of the model, as well as a sensitivity analysis. In addition to the *Honda-ẟ* measure that we used in the original submission to measure the extent to which emergent fields were Dirichletform, we have in the Revision devised additional metrics of the pattern quality that allow us to better quantify the correspondence between real and simulated barrels as the parameters are varied (in terms of barrel area and map topology; see final section of Materials and methods). Using these metrics, we analyse the results of 216 independent simulations, presented in the new Figure 2, as well as 200 new simulations used in the new Figure 3 to present an analysis of the sensitivity of the model to perturbations to the following inputs: (i) the magnitude and offset of the noise applied to a_i_ at t = 0, (ii) random noise applied to the interaction parameters, γ, (iii) random noise applied to the guidance fields, and (iv) the angle and magnitude of one of the guidance fields.

Based on these new analyses we arrive at the following prediction about how the tension of competition between thalamocortical axons and their lateral branching (diffusion) manifests in barrel organization:

“First we observe that self-organization is highly robust to the ratio ɑ/𝛽, across five orders of magnitude, with respect to all three metrics. […] Together these results indicate that when competition is strong, the rate of diffusion determines a trade-off such that fields emerge to be barrel-shaped when diffusion is fast and they emerge to be more selectively innervated when diffusion is slow.”

We now include several examples of patterns that are predicted to emerge in different regions of the parameter space. Specifically we confirm in simulation an intuition about the influence of cortical sheet size that we articulated only informally in the original submission (see new Figure 2B; specifically plot iii):

“The parameters of the example simulation are indicated in Figure 2A using an asterisk. […] In this context, it is interesting to note that for small D, the organization is predicted to be topological but highly irregular, with a general expansion in the territory occupied by the central versus peripheral domains that would presumably manifest as an absence of identifiable barrel fields (Figure 2B iii).”

We also demonstrate in the revision how the model can be used to study the effects of specific experimental manipulations that have been conducted with the whisker-barrel system, by replicating two whisker deprivation experiments:

“Finally, to investigate the response of the model to environmental manipulation, we simulated a whisker deprivation experiment. […] Together with the results of simulated misexpression, the consistency of the simulated whisker trimming results with those of the original studies demonstrates how the model can be used to investigate the contribution of intrinsic and extrinsic factors to the development of cortical fields.”

Reviewer #1:This compact paper proposes a self-organization model for formation of whisker barrels. The key idea is that reaction-diffusion dynamics can lead to the observed topology, in the absence of pre-defined centers for the barrels.The model is well presented and the motivation of mathematical choices is mostly clear. It may be worth expanding on the motivation for competition for axonal branching (Equations 3 and 4).

Thank you. We have expanded on the motivation for competition for axonal branching as outlined in our responses to point 1 above.

It is a little unclear how the misexpression experiment (Shimogori and Grove, 2005) in Figure 1E was done. The simulation approach and outcome for this section is described very tersely.

We have expanded our description of this experiment in the main text, as follows:

“To further investigate the interplay of genes intrinsic to the developing neocortex and extrinsic factors such as thalamocortical input, we simulated two well-known experimental manipulations of barrel development. […] The result after 30000 iterations, and otherwise using the parameters of the example simulation, was two mirror-symmetrical barrel fields comprising 2𝑁 barrels (Figure 4B), consistent with the outcome of the original experiments.”

The authors also mention another easily modeled experiment, in which capybara brains lack barrels because they are big. It should be a simple matter to do this run.

Thank you for the suggestion. As explained in response to point 3 above, we have run this simulation as part of the parameter sweep, and the results (as anticipated in the original submission) are shown in the new Figure 2Biii.

Overall I feel this study presents an attractive and compact model for the formation of whisker barrels, which has good biological motivation, and does a good job of reducing assumptions and molecular guidance cues.Reviewer #2:This is an interesting paper that with a few assumptions shows that an old model for areal formation in cortex is sufficient to quantitatively reproduce the patterns of barrels observed in mouse S1. It would appear from the model that the key is the parameters γ_ij_ that are presumably (hypothesized) to be assigned at the level of the thalamus. I have a few questions about the paper.1) Does the same model work with respect to projections from the brain stem (barrelettes) to the thalamus (barreloids)? This would be a good way to check the ideas. Related to this, is it true that the barrelettes (barreloids) precede the development of the barreloids (barrels)? It would seem to be necessary? Or perhaps, starting with a double gradient in the thalamus and cortex and a prepattern in the barelettes, would the correct patterns emerge simultaneously?

To answer the reviewers questions directly – yes, our thinking is that this model could be applicable for thalamic pattern formation, though we note that there is a more intricate 3D (pepper-shaped) structure to the thalamic barreloids that would also need to be considered in an extended version of the model, and there is less data available that describes these structures insofar as would be necessary to validate such a model. Yes, the formation of discrete whisker-related structures (or at least their visibility using current experimental techniques) precedes from brainstem barrelettes to thalamic barreloid to cortical barrels, though the (undifferentiated) projections from brainstem to thalamus to cortex are in place just before the barelettes become apparent. We are keen in future work to extend this modelling framework to investigate pattern formation along the neuraxis.

2) There seems to be a strong prediction in this concerning the development of the patterns over time. Figure 1C indicates that early on there are large distortions in the shape of the barrels particularly in Figure 1D, E rows. is this known to occur?

Care should be taken when relating the time-evolution of the simulation to the time-evolution of the development of barrels. The model considers axon branching density and connection density in the cortex without explicit consideration of which layers these quantities may be evolving within. It is known (e.g., from Agmon, 2005) that the axons initially form an undifferentiated mass in the subplate, and it appears to be here that they find their location before growing up, fairly straight, into layer IV where they then form the barrels. Thus, distortions in the pattern seen early on in the model development may be observable in the subplate, without necessarily being observable in the barrel field, and we are not aware of an appropriate dataset for subplate pattern formation that would be applicable.

3) It seems to me that without the chi, then possible connections plus axons are conserved which is reasonable. But with the necessary competition, there seems to be a flaw in the model if they have to renormalize at each point. If axons make connections should they not be lost from the pool forever (this is the -dci/dt the model). For example, since the gradient has no flux in the original K&E model, there is conservation of the total number of connections and axons of a given type. (int ai+ci dx = constant). This principle seems to make sense to me. However, the competition term χ_i_ seems to disrupt this. Is there a way to introduce the axonal competition in a way the prevents the unrealistic (or biologically implausible, at least) renormalization at each step? I'd be more comfortable with the model if there were a more physiological way to renormalize. For example, I don’t know if the authors considered something like an additional flux of the form:Xai∇1N−1∑j≠iajThis makes the axons of type i move away from type j while at the same time enforcing conservation without recourse to some sort of post normalization.

We thank reviewer 2 for this very insightful comment. We have addressed this comment in our response to point 2 above. We now present the model with the competition expressed in the form suggested, and we certainly agree that the model is much more elegant in this form.

Reviewer #3:The manuscript studies a theoretical model within the framework of reaction-diffusion equations coupled to signalling gradients to possibly explain the emergence of whisker barrels in the cortex.1) The model considered by the authors is identical to the one studied by Karbowski and Ermentrout, 2004. The only new features are the extension of the original 1D model to 2D and the addition of an extra term to represent competition in axonal branching.

This is correct. Though note that in the original simulations of Karbowksi and Ermentrout, 2004, a given field becomes localised around a specific point that is specified by a unique combination of two (or more) guidance fields, whereas in our simulations we show that the locations of many fields can be reliably influenced by only two fields.

2) The authors consider 2 guiding fields. What are their explicit spatial profiles? Notice that since these fields essentially guide the emergent pattern and hence their profiles, in relation to the geometry of the 2D domain, are crucial. A different profile would certainly lead to a different pattern. I feel that it is not enough to say '…linear signalling gradients aligned with the anterior-posterior and medial-lateral axes….' since the domain is 2D and of non-rectangular shape.

Minor edits to the Introduction text underscore that gradients in the guidance molecules are not restricted to the barrel field domain but extend across the entire cortical sheet (e.g., see Miller et al., 2006, Figure 11, for a clear illustration), and in the Results we have clarified:

“These gradients are shown with the barrel field boundary in Figure 1A for clarity, though like ephrin-5 they may be thought of as extending across the cortical hemisphere (Miller et al., 2006)”.

Note that the new Figure 3, which accompanies the sensitivity analysis, includes an analysis of the sensitivity of the model to various perturbations to the profile of the chemical cues, and that the specific profile is clarified in the corresponding caption:

“The effect of setting the rotational angle of the linearly varying guidance field, 𝜌1 (x), to 𝜙𝜌1, and modifying its overall gain to 𝐺𝜌1, whilst keeping the parameters of 𝜌2(x) unchanged from those used in the example simulation, for which 𝜙𝜌2 = 84°and 𝐺𝜌2 = 1.”

Note also that in the revision we have included an analysis of how the relationship between the two guidance fields affects pattern formation (new Results section on sensitivity analysis and new Figure 3A).

3) The justification for the introduction of the extra term for competition amongst axons (Equation 3) is missing. Why that form? What is the reasoning for introducing axonal competition? What essential features of the resultant patterns are missed out if this term is absent? Or has a different form? In the Discussion section, the authors mention, without any justification, that the conservation of branch density in each projection is a key requirement for the emergence of barrel patterns. This is totally unclear.

This comment is addressed through our response to point 2 above.

4) Related to the above point, the authors mention that the axonal branch density is bounded by their dynamics. I presume that the integrations on the RHS of Equation 4 are spatial integrals over the domain. Then how come a spatial index survives in the LHS of this equation? How did the authors arrive at this equation? Is there a continuous-time version of this equation (like a conservation law), i.e., one that does not make a reference to the discrete time-stepping dynamics?

We understand and appreciate the points raised here, as they continue on from the previous point. As explained in full in our response to point 2 above, the post normalization mechanism (original Equation 4) has indeed been replaced by a continuous-time version (revised Equation 3).

5) A typical mathematical modelling study should explore the space of relevant parameters to demonstrate the possible range of behaviours that the model can exhibit. This is usually presented as a phase-diagram. The authors do not explore the parameter space (or the possible spatial profiles of the guiding fields) in their study.

Thank you for encouraging us to fully explore the parameter space. This comment is addressed in response to point 3 above. Briefly, we have included results from a full parameter sweep (new Figure 2), and also include as part of a sensitivity analysis (new Figure 3) an investigation of how the profiles of the guiding fields affect pattern formation.

6) Throughout, the authors emphasize the spatial-locality of their mathematical model and conclude 'Hence the simulations demonstrate how a self-organizing system…'. A mathematical model with spatial-locality alone does not imply self-organized dynamics. With a sufficiently large number of spatio-temporal fields (N=42), and the concomitant parameters, and non-autonomous guiding fields, it is possible to reproduce any desired pattern. As such, it is crucial in the mathematical modelling of living systems to delineate the essential requirements from the incidental.

We agree that a mathematical model with spatial locality alone does not imply self-organized dynamics. However, nothing in the formulation of the model locally at the resolution of cell-cell interactions was designed explicitly (in terms of any global supervisory mechanism present in the model) to produce the key aspects of the emergent patterns as they may be described at the global scale (i.e., a Voronoi tesselation of fields that each comprise hundreds of cells). The guidance gradients (and the overall boundary shape) are indeed defined globally, but these gradients vary linearly across the domain, and they do not directly specify the Dirichletform shapes that emerge from cell-cell interactions. We trust that this point is clearer in the light of the results of the parameter sweep and sensitivity analysis, which demonstrate that the form of the patterns that develop in simulation is highly robust to variation in any of the component parameters.